# Adopt Big-Data Analytics to Explore and Exploit the New Value for Service Innovation

**Nopsaran Thuethongchai [1],* , Tatri Taiphapoon [2], Achara Chandrachai [3] and Sipat Triukose [4]**

[1]  Technopreneurship and Innovation Management Program, Chulalongkorn University,
    Bangkok 10330, Thailand
[2]  Faculty of Communication Arts, Chulalongkorn University, Bangkok 10330, Thailand; Tatri.t@chula.ac.th
[3]  Chulalongkorn Business School, Chulalongkorn University, Bangkok 10330, Thailand;
    achandrachai@gmail.com
[4]  Research Group on Applied Computer Engineering Technology for Medicine and Healthcare, Big-data
    Analytics and IoT Center (CUBIC), Chulalongkorn University, Bangkok 10330, Thailand; Sipat.T@chula.ac.th
*   Correspondence: nopsaran@gmail.com; Tel.: +66-9-5956-5156

**Abstract:** Big-data analytics is gaining substantial attention due to its contribution to the process of determining business strategy and providing valuable information for the design and development of service innovation. The principal objective of this research is to study the adoption of big-data analytics for service innovation. The focus will be on leveraging features of data analytics to capture genuine customer's requirements from the communication data through the digital service channel. This study used mixed methods research of documentary research, with supplementary semi-structured interviews. The interviews were conducted with 11 executive managements who have more than ten years of experience in data analytics or service development. The result of the research found that organizations in the services industry are using big data analytics to build capabilities to gain competitive advantages as well as the ability to rapidly and accurately respond to the market's demands. The process of adopting big-data analytics for service innovation described in this article consists of seven essential procedural steps that impact the success of the development of service innovation, and also considered with the objective of increasing effectiveness in opportunity identification and reduce complexity in the fuzzy frond-end service innovation development theory.

**Keywords:** opportunity identification; service innovation; new service development; big-data analytics; data analytics

## 1. Introduction

Service Innovation is the most important factor for supporting the service business sector in creating value for their customers and organization to survive and grow sustainably. For decades, the service business sector has been transformed from a product-focused business to a service-driven economy. Service is an essential factor in the global economy. More than half of GDP values for most developed countries are aimed at the Service sector, and it was estimated that the economy and employment rate growth in the 21st century would be dominated by service (Woodward 2009). As an overview of the service business sector, the data service business has grown by leaps and bounds, for example, financial service business, telecommunication business, e-commerce business service, and professional service (Karmarkar and Apte 2007). For these reasons, many organizations point their focus on innovation development, which requires new service development (NSD) processes and procedures following current customer behaviors and expectations (Johnson 2000).

In Thailand, the service sector plays an important role in country economic growth, which has the most significant GDP portion, or more than half of the total GDP, and highly increases the country's

employment rate. However, Thailand's GDP has a sluggish growth rate compared to those countries with 70–80% development in the service sector. For Thailand, the percentage of the economic value of the service sector is 50% despite 49% of employment rate, which is contributed from traditional service relying on low-skilled labor, less technology process, and domestic market dependence, from Service Production Index SPI 2017.

Considering innovation capability by The World Intellectual Property Organization (WIPO), a specialized agency of United Nations (UN), in Global Innovation Index (GII) 2019 from 129 ranking countries, Thailand has been ranked as the 43rd, which takes one step higher (44th) than the 2018 ranking index. This represents the effort of using innovation for country development (Dutta et al. 2018; Global Innovation Index 2019). One of the main factors impacting innovation development is from the rapid improvement of technology and ICT adaptation leading the transformation of customer behaviors and ways of doing business nowadays. Besides, customers play a variety of crucial roles and are able to conduct self-study or contact service providers through the digital channel, which is different from before (Christen 2012). The latest research shows the increasing number of methods and channels for customers to connect to an organization, while numbers of traditional ones continuously decrease (Christen 2012).

The service innovation process is out of focus or is being less focused on by most of the Innovation Management in the service sector in Thailand. As a result, systematic new service development methods are not widely applied. It is, therefore, urgent and critical to create and manage service innovation for creating value and increasing competitiveness for the growth of the Thailand service sector and effectiveness in conducting business. Besides, adopting data analytics through digital channel, which is massively available with innovation development plan, is essential for meeting the needs and rapid change called "fuzzy front-end"; it is in a disordered state of initial project period, and it is critical period for opportunity identification, as the innovation development process will be unsuccessful if new service development (NSD) is mistakenly proceeding, resulting in loss in both budget and other resources.

This research aims to formulate a service innovation process by adopting big-data analytics according to the advancement of IT nowadays, towards new service development (NSD) for the benefits and effectiveness of empirical research. This research uses mixed-method, starting from documentary research as a secondary study, together with semi-structured interviews as a primary study by applying qualitative method, which received acceptance as research method (Strauss and Corbin 1998) by interviewing 11 executive managements with successful experiences in establishing service innovation and being specialized in service innovation development for more than ten years. The main research questions are:

- How to conduct a big-data analytics process, needs exploration, and opportunity identification for service innovation development to identify the opportunities that meet the needs and are promptly available in current fast-moving circumstances.
- What are the success factors for adopting big-data analytics in service innovation development, and what are the limitations?

## 2. Materials and Methods

### 2.1. Theoretical Framework

#### 2.1.1. Sources of Service Sector Growth

The growth of service sector around the world that is derived from IT advancement, changes in customer behavior creates new demands, and most importantly, the advancement impacts digital service growth as well as focuses on data service, which is growing tremendously, e.g., connecting internet through iPhone and using internet for mobile banking services such as money deposit, withdrawal, transfer, or e-banking account summary check (Karmarkar and Apte 2007; Scarpetta et al. 2000).

The advancement of technology, IT, and telecommunication causes the rapid growth of the service industry up to a global scale. This accelerates pressure to service providers for new service developments (Menor et al. 2002). As a consequence, there are advantages from service completion in offering new services; for instance, (1) increasing profit from existing offerings, (2) attracting new customers, (3) strengthening the loyalty of existing customers, and (4) enlarging market opportunity, according to the recent report, shows 24.1% of income revenue of service company generated from new services launched during the past five years, and this turns out to be 21.7% of company profit (Griffin 1997).

The service industry can attract more than three-fourths of foreign direct investment in the economy, where it has the highest rate in employment and new job opportunity, in addition to adding new service into the global economy, e.g., legal service, engineering service, other professional services, computer service, and telecommunication service (OECD). It is, therefore, unavoidable for the service industry to come up with new services and to adapt correspondingly to current customer behavior. There are many forms of digital services, such as Facebook, Chat, Website, and Social Media, that continuously increase to approach customers, including future business diversification plan of service innovation in order to drive an organization to get through and grow competitively. Miles (2008) It has been noticed that now to the future, the growth of the Information economy soars rapidly by using information from data that are relevant, accurate, timely, and concise, depending on receivers and senders. Practically, information is generated by data processing from systems or machines, and by humans.

The service sector providing data service tends to grow; meanwhile, the physical product sector gradually declines (Karmarkar and Apte 2007). Furthermore, interactivity has higher growth than other sectors. Plus, there is substantial demand for businesses providing data and communication between provider and user, which has a high volume of data flow across both sides, leading to an upsurge of high standard IT adaptation; in the financial field, for example, (Miles 2008). Additionally, the more communication customers have, the more products and services designed to serve their demands will be developed, by using big-data analytics process leading product and service development for individual needs (Kotler et al. 2016).

2.1.2. Innovation in Services and New Service Development Theory

Definition and format of service innovation: Innovation means the success of optimizing new ideas at its highest benefit (Francis and Bessant 2005). Furthermore, according to National Innovation Agency, Thailand's public organization, which supports and drives innovation development, determines that innovation is the novelty derived from knowledge and creativity, which is beneficial to economics and society. Innovation is the process of integrating knowledge, creativity, and management for innovation businesses or new businesses initiating new investments to boost country competitiveness. Typically, successful innovation is created from multiple aspects. Fundamentally, Innovation development requires various fields of knowledge, internal collaboration, and effective communication, which broaden perspectives for innovation development (Mabogunje et al. 2013).

Miles (2008) defines service innovation as a new service demonstration or existing service innovation development for a considerably better advantage. Regularly, new service development (NSD) is created from interaction between customer and service agent, who is able to identify the actual demand (Unmet Needs); moreover, service innovation is shaped from the findings of existing data or analyzing data which indicates the customer's needs, helps to understand the customer and adopt for new service innovation, or effectively value-adding for existing service improvement (Fitzsimmons et al. 2008). Frequently, innovation is perceived as creating something new, and is the absolute outcome of service. It is unnecessary to be a new service product; on the other hand, to modify the degree of existing service. There are two classifications of Innovation: (1) Radical Innovations, which are completely new services offering or new delivery management for existing services, (2)

Incremental Innovations, which is to renovate current service for more significant enhancement (Fitzsimmons et al. 2008).

New service development (NSD) is the process and series of formal activities indicating the progression from new creativity until selling service into the market (Scheuing and Johnson 1989). Definition of Service Creation is "methods consisting of activity series for developing or renovate service innovation". What are the differences between Service Design and New Service Development? Schneider and Bowen (1984) stated that design is the initial step for the new service development process. They noticed that service design would indicate fundamental structure and context, including service strategy (Roth and III 1995; Roth and Velde 1991). Meanwhile, new service development (NSD) is the overall process for developing new services. The design problem is critical for new service development (NSD) process determination. Bowers (1989), based on Edvardsson and Olsson (1996), they insisted on dividing new service development into three activities; (1) Service Concept Development, (2) Service System Development, (3) Service Process Development. The service concept is an explanation of a customer's needs and satisfaction. Service System is a necessary resource that is necessary for those services, comprising company employees who provide physical and technical services, an organization as a base structure, supporting system management, and the customer as the "co-producer."

The service process is the chain of activity for service to be executed. Scheuing and Johnson (1989) affirmed that new service development (NSD) does not happen by chance; however, it is initiated from the design process, and systematic execution found from an integrative demonstration of Johnson (2000), which combines all previous researches in new service development process with four main methods: (1) Design, (2) Analyze, (3) Develop, (4) Launch and Feedback for new service development (NSD) process loop. Frontend offer, especially strategy development or concept of intangible service design analysis, are difficult. There was an essential recommendation for the new service development (NSD) process cycle, which was different from the new service development process for the reason that it is a non-linear process and emphasized "Enablers", which are working team, organizational culture, accommodative designing tools, or new.

### 2.1.3. Adopt Big-Data analytics in Service Innovation Development

De Mauro et al. (2016) defined big-data as a massive number of substantial information with assorted variety that rapidly increases in a quantity, which requires technology to run the analysis for converting them into meaningful data for business purpose. Laney (2001) is one of the first persons who identified three main challenging characteristics of data to be managed, which are (3V) massive volume, velocity, and variety. These 3V are widely accepted as the characters of big-data, which later on, there is more explanation added; for instance, Veracity by IBM, Variability by SAS, Value by Oracle, and Visualization.

Using big-data for its benefits is required for the practical process of the analysis and management for obtaining meaningful data for business purposes. Labrinidis and Jagadish (2012) suggested the five processes; 3 of them are data management, and 2 of them are data analysis. The processes are data management; (1) Acquisition and Recording, (2) Extract Cleaning and Annotation, (3) Integration, Aggregation, Representation, and Data Analytics, (4) Modeling and Analysis, (5) Interpretation. Comply with Wirth and Hipp (2000), specifying CRISP-DM: Cross-Industry Standard Process for Data Mining with 6 Step. The steps are (1) Business Understanding, (2) Data Understanding, (3) Data Preparation, (4) Modeling, (5) Evaluation, and (6) Deployment

Data mining is a multi-disciplinary technique including Artificial intelligence, Machine learning, Statistic, Database system (Kantardzic 2011); the critical objective for conducting data mining is to extract the advantageous information and knowledge from a substantial number of raw data presenting in many formats; Classification, Estimation, Clustering, Associative rules. Additionally, there is sub-major knowledge under data mining, which is text mining for understanding the unstructured data, such as those dialogues through Social network, Email, Chat, Call Centers, or Websites. Conducting information extraction is to understand unstructured data and convert them into structured data,

including sentiment analysis for opinion mining, and perform text summarization for the summary of essential information.

Venkatram and Geetha (2017) stated that text mining is one of the text analytic processes for insight that resides in dialogue from E-mail, Conversation, Feedback, or other sources of information from the Websites and Social Network. Performing unstructured data mining using natural language is required for statistic methodology and machine learning to comprehend due to the unstructured form of communication, specific vocabulary, and variety of messages depending on the characters of those who communicate. In other aspects, it was indicated that using data analytics as one of the components for service innovation can design and deliver new valuable services to the customer for a company to be differentiated and having competitive advantages. Currently, technology is extensively used in communication, financial activity, and providing service through digital channels, resulting in much information being transferred and recorded on an information technology system, in which organizations can use this massive amount of data to analyze by using technology. The process of the analytic data system has progressed rapidly for design and creates new strategies in providing service to the customer; however, there is no research found to identify process or success in this matter clearly. Lehrer et al. (2018) researched big-data analytics for service innovation for an organization consisting of 2 components:

1. Using qualification and core competency of technology to conduct big-data analytics for service process automation for trigger-based and preference-based service operation.
2. Designing a service process by using technology as a part of the service process (IT-enabled service process), which there is the interaction between a service agent and technology. Big-data analytics for interaction with the customer by providing trigger-based service and preference-based service. The service industry did not focus on the context of using big-data for the study of new service development, and yet focused on the context of using data as a key for value creation services. Knowledge development for designing services is still in need of service improvement and value creation. Knowledge in designing process should be duplicated and be evaluated for the completion of process improvement.

Looking at the benefits point of view of big-data impacting service development, it found issues in data management behind its benefits (Lim et al. 2018). Big-data analytics process for service innovation development can be summarized in the process from a literature review in Table 1, and information to adapt to the service innovation process in Figure 1.

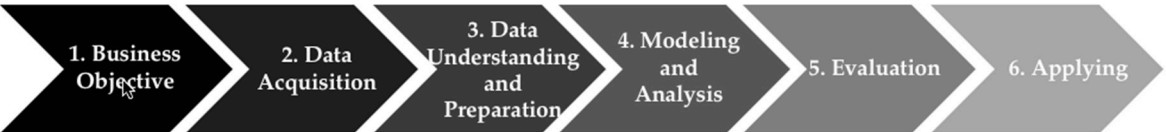

**Figure 1.** Propose big-data analytics process for service innovation development. Source: Compiled by authors according to documentary research.

**Table 1.** Big-data analytics process for service innovation development.

| Year | Authors | Step 1 | Step 2 | Step 3 | Step 4 | Step 5 | Step 6 |
|---|---|---|---|---|---|---|---|
| **2000** | Wirth, R., & Hipp, J. (2000, April) | Business Understanding | Data Understanding | Data preparation | Modeling | Evaluation | Deployment |
| | Wirth, R., & Hipp, J. (2000, April). CRISP-DM: Towards a standard process model for data mining. In Proceedings of the 4th international conference on the practical applications of knowledge discovery and data mining (pp. 29–39). London, UK: Springer-Verlag | | | | | | |
| **2004** | Berry, M. J., & Linoff, G. S. (Berry and Linoff 2004) | Identifying the problem | Transforming data into information | | Taking action | Measuring the outcome | - |
| | Berry, M. J., & Linoff, G. S. (2004). Data mining techniques: for marketing, sales, and customer relationship management. John Wiley & Sons. | | | | | | |
| **2011** | Sahu, H., Shrma, S., & Gondhalakar, S. (Sahu et al. 2011) | - | Data preparation | Data mining | Information expression | Analysis and decision making | - |
| | Sahu, H., Shrma, S., & Gondhalakar, S. (2011). A brief overview on data mining survey. International Journal of Computer Technology and Electronics Engineering | | | | | | |
| **2012** | Labrinidis & Jagadish (2012) | - | data acquisition | data collection/extraction | data analysis | - | - |
| | Labrinidis, A., & Jagadish, H. V. (2012). Challenges and opportunities with big data. Proceedings of the VLDB Endowment, 5(12), 2032–2033. | | | | | | |
| **2013** | Sumathy, K. L., & Chidambaram, M. (2013) | - | Gather information/Data processing | Attribute Generation and selection | Mining Pattern with apply difference techniques | Evaluation | - |
| | Sumathy, K. L., & Chidambaram, M. (2013). Text mining: concepts, applications, tools and issues-an overview. International Journal of Computer Applications, 80(4). | | | | | | |
| **2014** | Gandomi & Haider, (2014) | - | Data Acquisition/Extract/Cleaning and Annotation | Data Integration, Data Aggregation and Representation | Modeling and Analysis | Interpreta- tion | - |
| | Gandomi, A., & Haider, M. (2015). Beyond the hype: Big data concepts, methods, and analytics. International journal of information management, 35(2), 137–144. | | | | | | |
| **2014** | Dang, S., & Ahmad, P. H. (2014) | - | Collecting/ Convert this information received into structured data | Identify the pattern from structured data | Analyze the pattern | Extract the valuable information and store in the Data base. | - |
| | Dang, S., & Ahmad, P. H. (2014). Text mining: techniques and its application. International Journal of Engineering & Technology Innovations, 1(4), 866–2348. | | | | | | |
| **2018** | Lehrer (2018) | - | Data Sourcing /Storage | - | Event recognition and Prediction | Behavior recognition and Prediction/Rule bases analysis | Visualization |
| | Lehrer, C., Wieneke, A., Vom Brocke, J. A. N., Jung, R., & Seidel, S. (2018). How big data analytics enables service innovation: materiality, affordance, and the | | | | | | |
| **2018** | Lim, C., Kim, M. J., Kim, K. H., Kim, K. J., & Maglio, P. P. (2018) | - | - | data collection | data analytics/information creation | Information delivery design | - |
| | Lim, C., Kim, M. J., Kim, K. H., Kim, K. J., & Maglio, P. P. (2018). Using data to advance service: managerial issues and theoretical implications from action research. Journal of Service Theory and Practice, 28(1), 99–128. | | | | | | |

Source: Compiled by authors according to documentary research.

*2.2. Research Methodology*

This research study is using mixed method starting from documentary research, which is secondary data, together with a semi-structured interview, which is primary data, by using a qualitative method which received acceptance as research method (Strauss and Corbin 1998) under the research approved on ethics review by a group of Human Ethics committee (Chulalongkorn University).

The scope of this research is to study the adoption of big-data analytics in service innovation, focusing on the process and initial procedure of needs and opportunity identification, which is fuzzy front-end of new service innovation for the topic, corresponding to digital service according to the research framework shown in Figure 2.

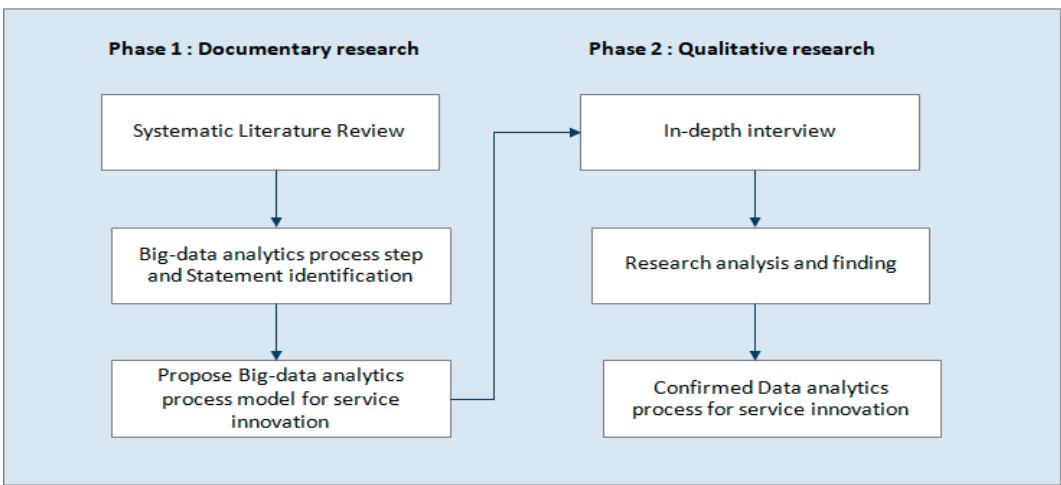

**Figure 2.** Research process. Source: Developed research framework by authors.

2.2.1. Documentary Research

Documentary research is the second study starting from literature review, theory, or related philosophy during the year 2000 to the present, and in the case of before the year 2000, there is relevant research that is significant to this study, and it will be included to cover all research context.

Sixty-five studies have keywords in research topics, abstract, and academic database keyword Scopus, Science direct. Dividing topics into; conceptual theory of service innovation, new service development, big-data analytics adoption to analyze, summarize, and link the related important factors to research questions, and to create tools for semi-structured interviews to evaluate, define the variable details and process to adopt big-data analytics for service innovation to have a framework of initial questions for semi-structured interviews in order to have the concept and summary of big-data analytics adaptation in the proper process of service innovation development.

2.2.2. Qualitative Research (Semi-Structured Interview)

Collecting data from documentary research for feedback and variable detail are the basic framework questions for a semi-structured interview in finding big-data analytics adaptation process for service innovation development, especially through a digital channel, by interviewing 11 executives from mid-level to senior who have more than 10-year experience in service innovation development and being service innovation development specialists.

2.2.3. Source of Research and the Selection of Informants

The sampling selection is determined from the service business trend that emphasized providing data as their services (Karmarkar and Apte 2007), including Telecommunications, Financial Service, Professional Services, Broadcast Services, and Education.

The researcher performs nonprobability sampling randomly for informants by using a snowball technique to select senior-level executives and experts for answering research issues, and suggests the next informant whose qualifications met the target group. The interview began with a contact center management executive in a top bank in Thailand, which has the highest income in Thailand and meets all criteria of relevant qualification, including (1) being in the large organization providing the number 1 in the financial service business in Thailand, which generates the highest revenue in the year 2019. (2) Having 23 years of experience in service innovation development. (3) Their organizations have approximately 20,000 employees. (4) Their organizations serve approximately ten million customers. (5) Their organizations provide a channel for digital services covering all channels, comprised digital channel service center, Chat, Line, Facebook, Email, Pantip.

The informant was requested to refer a list of names and their contact details of the next informants, who were the executives or service innovation development specialists with more than 10-year experience. This is due to the necessity of the various angles of knowledge from informants to identify obstacles and factors impacting success. The interview was conducted until the data collection reaches its stable level, and there are similarities in the answers of those from previous informants. Additionally, the interview was ended when the informant was not able to further refer to others due to qualification mismatch.

There are 11 informants for this research, and the interview has been started since September 2019 till December 2019 in Bangkok, Thailand, consisting of the following business service segment: (1) Financial Service, (2) Telecommunication Service, (3) Insurance Service, (4) Electronic Commerce Service, (5) Logistics Service, (6) Software Development Service, (7) Service-related Data analytics Service. The informant information shown in Table 2.

**Table 2.** Informant Information.

| Service Segment | Position | Management Track | | | Experience (Year) | |
|---|---|---|---|---|---|---|
| | | Middle Management | Top Management | Specialist Track | All | In Service |
| Telecommunication Service | Senior Manager | | | X | 30 | 20 |
| Software Development Service | Managing Director | | X | | 28 | 28 |
| Telecommunication Service | Associate Director | X | | | 28 | 15 |
| Insurance services | Director | | | X | 26 | 10 |
| Electronic Commerce Services | Senior Director | | X | | 25 | 15 |
| Financial services | Senior Advisor | | X | | 23 | 23 |
| Insurance services | Senior Vice President | | X | | 23 | 10 |
| Logistic Service | Senior Manager | X | | | 18 | 12 |
| Consumer Discretionary | Director | | X | | 17 | 10 |
| Marketing & Insurance | Chief Executive Officer | | X | | 15 | 11 |
| Telecommunication Service Provider | Vice President | | | X | 15 | 10 |

Source: Compiled by authors according to in depth-interview.

*2.3. Research Tools*

Design semi-structured interviews from documentary research development, which complied with the introductory concepts and findings to determine the questions of adapting big-data analytics for service innovation development in order to provide a comprehensive framework. In addition to autonomous condition to the informant, the open-ended questions were used. Moreover, there was a validity test to check the tools by experts and scholars before using it.

*2.4. Collecting Data*

The rights of the sampling group were protected when collecting data using semi-structured interviews from key informants. At the beginning of interview session, there was a self-introduction of the researcher, followed by a verbal request for the permission from the informant and an explanation of the objectives of the interview, plus the duration of the interview session, which is around 1.5–2 h, with seeking permission to note down and audio record the content for review later. In order to check for accuracy, the researcher summarized the process according to the information from an informant, including other factors, immediately after the interview for the informant to review and confirm the conclusion. In order to protect the privacy rights of the informant, the unique code is set for summarizing the conclusion and analyzing data for the report. There was data verification to ensure the liability and accuracy of data with triangulation methodology (Cohen et al. 1994) generated from various sources, comprising three subjects:

1.  A semi-structured interview with ten management executives or experts from various service industries who have more than 10-year experience in managing and service innovation development.
2.  Testing by having face validity in every interview, and request for clarity and examples for any unclear issues.
3.  Conduct content analysis on service innovation development within informants' organization.

Issues found in the data collection process is due to interviewing executive managements and experts, which is time-consuming for planning for data collection, since having to send the permission request letter for conducting the interview, schedule an interview meeting, which some rejected due to schedule conflict or some informants require more than one month in advance for scheduling the interview, or some rejected as the question is quite specific or confidential.

## 3. Results

Qualitative research used a mixed-method from conducting documentary research to semi-structured interviews to evaluate and explore the process for service innovation development. The researcher used the interview data from both note-taking and audio recordings for analyzing by typological analysis by classifying data from keywords and domain analysis, grouping words with similar meaning into keywords, and using taxonomy analysis for finding the relationship between words and keywords in the overall picture. That includes content analysis and constant comparison on information acquired from the interview and observation for more clarity (Glaser 1965) summary of results and opinion of key formants on the two essential research questions are:

*3.1. How to Conduct a Big-data analytics Process, Needs Exploration, and Opportunity Identification for Service Innovation Development to Identify the Opportunities that Meet the Needs and Are Promptly Available in Current Fast-Moving Circumstances*

3.1.1. Adapt Big-Data Analytics Process for Service Innovation

A semi-structured interview from all key informants can summarize the process of adopting big-data analytics for service development, according to Table 3.

**Table 3.** Process prioritization for big-data analytics for service innovation.

| Recommendations by Key Informants | Number of Informants Agreed | Informant Number |
|---|---|---|
| Business Objective and Requirement | 6 | 03, 05, 06, 09, 10, 11 |
| Data Acquisition | 11 | All |
| Data Understanding and Preparation | 7 | 01, 02, 03, 04, 08, 09, 10 |
| Identifying the problem and customer insight | 11 | All |
| Analysis and Modeling | 11 | All |
| Evaluation | 8 | 01, 02, 05, 06, 07, 08, 10, 11 |
| Deployment and Apply | 11 | All |

Source: Compiled by authors according to a semi-structured interview.

Process for opportunity identification from adapt data analysis of customer interaction for new service development, leading to new service development (NSD) process. The informants mainly focus on using various technologies applying to data analysis and customer insight. To identify opportunities in the process of service innovation development, there are different opinions on the procedures. Considering an informant who is in direct service segment gave an opinion on opportunity identification that first process should determine value of identifying customers' and market needs and indicating source of data, which covers overall customer experience with organization through all channels for discovering for all needs, get feedback from customer directly, which is outside-in approach, and able to identify customer's pain–point. Consequently, collecting customers' conversation dialogue which was requesting service, inquiring for information, ordering the service, notifying service problems, including data from customer relationship management, is very beneficial information for the organization. It is, therefore, necessary to specify as the first process on how to gather data and ways to gather information from the voice of customers, which is different from defining requirements or identifying opportunities based on management perspective or business perspective solely, which probably causes weakness in meeting the needs of the market and customers of service development. Especially today, when customers have changed their behavior quite fast, resulting organizations regularly and closely monitored informants no. 01, 02, 04, 07, 08.

On the contrary, another group of informants from those organizations supporting leading service pointed out that, firstly, it should begin with business objective or business requirement starting from demands to identify business requirement, having a clear objective at the beginning to help define the information needed to collect data that meets business needs, resulting in the efficiency in cost and resources—furthermore, having opportunity identification for the clear objective of verification and what to explore according to informant no. 03, 05, 06, 09, 10, 11. There are different opinions on the starting point of data analysis in service innovation development besides that more than half of the informants support the approach of starting by proceeding with the business objective and requirement. The researcher considered this as the issue not to be overlooked, as there is no clear evidence from documentary research and the semi-structured interview. Additionally, from the interview, the researcher is able to identify the gap of documentary research in the process of the problem and customer insight identification, which informants stated this process is very important because it is a process of finding value and opportunity of service innovation development, to be comprehensive and consistent into seven steps, which are (1) Business Objective and Requirement, (2) Data Identification and Data Acquisition, (3) Data Understanding and Preparation, (4) Identifying the problem and customer insight, (5) Analysis and Modeling, (6) Evaluation, (7) Applying findings to the service development of the innovation process, according to Figure 3.

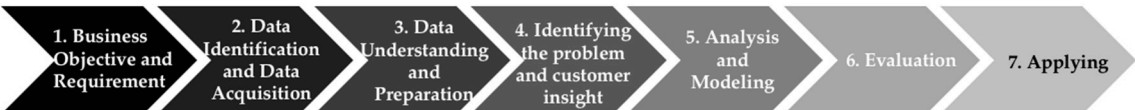

**Figure 3.** Process for adapt big-data analytics for service innovation. Source: Compiled by authors according to a semi-structured interview.

- Step 1: Business Objective and Requirement

It is to set objectives or goals for getting customer feedback or executing this project in order to have a clear understanding of the desired goal of the project or interested matters, and identify the objective, which is to determine opportunity identification for understanding the overall picture of the operation, which is not too broad or too narrow. This process is to verify and explore customer requirements and opportunity identification; therefore, if it is too narrow, it will cause a lack of identifying an opportunity, but to analyze data for those operation issues that already have clear answers.

- Step 2: Data Acquisition

In order to meet the business objective and requirement determined in step 1, it is essential to define the appropriate data input that will be used for analysis. The focus of this research is to design and develop new services for the customer, so data related to customer services, such as customer feedback (voice of customer) must be acquired. Since customer services are provided via a digital channel, communication data can be captured and stored in digital form. These unstructured communication data are mainly used as input for analysis and interpretation of customer requirements and customer's experiences from existing services.

- Step 3: Data Understanding and Preparation

The customer services related to data obtained from the various sources have to go through the data cleansing process. The data cleansing process detects and correct (or remove) corrupt, inaccurate data. This can be done manually or by using data cleansing tools that apply a pre-defined set of rules for data verification and correction. Data from various sources with different forms and formats have to be standardized and integrated. It is essential to have a clear understanding of data at this step. The correct data definition is crucial in data analysis. Furthermore, the quality of data must be examined. Experts in data analysis, who are familiar with the data, need to perform initial analysis to verify quality of the data and ensure that data is valid for further analysis.

- Step 4: Identifying the Problem and Customer Insight

At this step, valid and qualified customer service data obtained from the previous step undergo preliminary data analysis by the domain expert. The purpose of preliminary data analysis is to identify problems or issues that customers have encountered, as well as customer requirements on the products or services. The findings are then compared with existing products or services to identify the business gap, hidden or unknown demand of the customers. To achieve this, in-depth service comparison and differentiation with systematic problem root cause analysis methods are used by having experts brainstorming to generate ideas for solutions or services to solve existing problems. Then, idea validation is conducted with front line staff to determine the validity of the solution idea and determine any pitfalls that may occur.

- Step 5: Analysis and Modeling

The solutions or services ideals obtained from the previous step are a preliminary level and required in-depth detailed analysis. The further analysis aims to discover data patterns, customer behavior patterns, or hidden knowledge. The data analysis models or data algorithms are used to analyze large amounts of customer service data. The models used for analysis can be Data Association, Data Classification, Data Clustering, and Factor/Variable Effect Modeling. Many data analysis and statistical analysis tools are widely available for performing these complex data analyses.

- Step 6: Evaluation

The results of the data analysis using various methods must be evaluated to verify efficiency of the model as well as accuracy of the results. Mainly, the results of the analysis should address the

solutions to the problems and meet the business objectives. It should also be evaluated for its reliability by comparing the results with analysis by domain experts. The discrepancy is then used to adjust the variables or factors of the model to improve the accuracy of the analysis model. This step requires an expert level of data or statistical analysis.

- Step 7: Applying the results to service innovation

Taking knowledge extracted from big-data analytics to utilize in service innovation, which is considered as the front end of the new service development plan. That helps reduce confusion and ambiguity of the front end of service development, or called "fuzzy front-end of service innovation."

### 3.1.2. The Benefits of Adopting Big-Data Analytics in Service Innovation

From the interview found all informants commented that "adopt Big-data analytics is very beneficial for service innovation development" from 11 informants who have experience being experts in service innovation development for more than ten years, gave two main reasons as follows:

Firstly: All of the 11 informants stated correspondingly that analyzing data on customer interaction, which are massive and unstructured data, can help to have a better understanding for customer needs and customer insight and service request patterns which are unique, identifying individual customers' preferences. Furthermore, they can capture the shifting trends of customers' behaviors timely because of using big-data analysis through various channels with connecting the relationship to customer journey leading to have a more precise vision for overall customer experiences, unlike market surveys, which only some part of can be visualized.

Especially the organization, whose goal is to be number 1 in the field of services, they must use information related to customers with the data of customer interaction to analyze for both broad and in-depth understanding in order to improve the service or offer the new services to meet customer needs at the individual level. Apart from having an understanding of customer needs at the right time, an organization can specify whether or not they can serve customer needs, how much difference there is of current service offerings to what the customer requires, and to what extent of their needs, which is advantageous for using correct information received directly from customers.

The essential of key informants:

Informant no.01 mentioned that "apart from being able to understand what customers need, also we can comprehend the gap that company did not meet customers' needs, and how important those gaps were to the customers in order to prioritize issues to be fixed and things to be improved, to identify matters that customers value, and itss level of importance to facilitate the prioritization for things to be modified for improved." Informant no.02 added that "transaction information done by customers through various systems can be analyzed to understand customer behavior to have more understanding on what customers want, and to realize trends from customers' perspective of what has changed; moreover, it is accurate customers' data to identify right timing and period to serve customer's needs."

Similarly, Informant no.04 stated that "Currently, the company has many customer interaction transactions helping them to identify what products to be developed, which one is suitable, finding new prospects, and it is useful for better understanding the customer. Nowadays, they are using this information to improve their current interactions, and initiate new things for better customer experience, develop current offers for the most positive impression. Therefore, it is essential for those companies that wants to be number 1." Besides, informant no.05 commented that "Analyzing data from customer interaction is useful, which is to be able to understand the nature of customer request that is beneficial, because if we really understand the needs of the customers, we can improve the service to serve their needs as much as possible. The improvement means to come up with services which meet customers' needs".

Secondly: All eight informants stated that adopting data analytics, including analyzing a large amount and unstructured data of customer interaction, can develop service innovation, which is to develop existing services and develop new services to meet the needs of customers both current and

prospective. Analyzing by groups, classifications, and at the individual level can specify different needs to offer a personalized offering and identify current market gaps which are yet to be fulfilled, create opportunities to present new services are undisclosed demand in the market, creating current competitive advantage. Today, service is much different from before because customer's needs and their behaviors are shifting rapidly and differently. It is necessary for the service sector at present to develop and improve services regularly in order to maintain its base market and grow continuously.

The essential of key informants: Informant no.03 pointed that "analyzing of customer interaction data help to visualize specific personalized needs, which can be used to develop and design customize service according to the preferences of each customer for the reason that nowadays the way of service has been obviously changed so that apparently we are not able to provide general mass services anymore, as the unique needs of each customer are more different than before" Informant no.07 indicated that "utilizing the benefits of data analysis for collecting customer data from their inquiries or service usage via a digital channel in service innovation development. Alternatively, compatible technology can indicate the needs that best match the target customers".

Moreover, informant no.06 added that "with this process, they could assess the situations based on actual data rather than predictions or the sense in order to perform service innovation" Meanwhile, informant no.09 pointed out that "service innovation is time-consuming and costly. The issue has a new service that failed to meet customers' needs; although, it has gone through a market survey or market testing, resulting in harmful consumption of cost, time, and resources in the development process that does not generate positive results. That is because the process of conducting market research creates high discrepancies from assigning group samples, the insufficient number of samples, formulating questions, conducting fieldwork, until the final process, which is to analyze and explain the results. As a result, the new service development process (NSD) is not as successful as it should be"

### 3.1.3. Characteristics and Types of Data for Service Innovation

Most of the informants gave the same information classifying data into three types; 1st type, structured data from customer relationship management system (CRM), which have been recorded in the system after the service, including systems that have transactions via operation system in each industry or creating several transactions, visiting various pages on the website where there is visiting history record, data from this channel consists of the type of customer profile, product profile, transaction history, customer requested history, service requested history, including other customer behavior and voice of customer. 2nd type, unstructured data which are customer interaction dialogue with the employee, Social Media message from Facebook, Chat, or Website. Both of these data are useful for analysis in order to proceed into the process of new service development. 3rd type, data from market research, which is the additional information to type 1 & 2. Usually, in the past, we only use market research data to develop service innovation. Currently, it is not enough to identify the needs of a diverse market; therefore, it uses the supplement information only.

### 3.2. What Factors Impact the Success of Applying Big-Data Analytic for Service Innovation Development and Limitation to Execute?

All of the informants mentioned that adopting big-data analytics for service innovation can identify factors impacting success, which are dividing into five areas:

- Organizational Capabilities: Due to data analytics for the service innovation process, it is necessary to have the experts with the knowledge of data analytics, business, and technology, which cannot be found in one single person, but through teamwork and cross-functional working style for many aspects of understanding. Besides that, data analytics and interpretation are important in the information interpretation for the reason that understanding the information requires a domain expert who is a specialist in the field as well. It is another issue that organizations must pay attention to, as it will create an error to the following step from the fault interpretation. (Informant no. 01, 02, 03, 05, 06, 07, 08, 09, 11)

- Resource in task domain: Using data analysis technology requires investment in both software and hardware, it is necessary to plan, to list out the details, and to evaluate investment value. An organization should consider its resource capability in other to conduct data analytics by using outsourcing options at the early stage and gradually develop the knowledge within the organization. In the case of strategic analytics, an organization should execute internally. (Informant no. 01, 05, 07, 08, 11)

- Data Security and Data Privacy: Protecting data and conditions relating to data privacy is important for an organization to minimize future discrepancies and to encourage the transparency of their data management plus regular evaluation. (Informant no. 01, 07, 10, 11)

- Data Management and Innovation Management: Is very essential, starting from setting clear outside-in objective from customer's perspective, defining useful information for the organization and planning of data gathering, implementing based on legal framework, promoting team to its high potential in data utilization and analytics, cross-functional working, proper budget allocation and technology investment, and systematic innovation project management. Furthermore, data management is important for the data-driven organization to have a clear systematic process due to many sources of system, scattered information, and no standard format of data. Hence, it is necessary to have complete data management for ready to use for data analytics for the organizational benefits. Additionally, managing innovation within the organization, it requires continuous support from management for creating motivation within the organization for creativity and to support the innovative culture of the organization. (Informant no. 01, 02, 03, 04, 05, 06, 08, 10, 11)

- Managing expectations of high-level executives: Those who do not see the value of using data, because big-data analytics are time-consuming and investment is required to proceed, or sometimes it generates different knowledge from the mindset of management's experiences. Sometimes, a top-down decision may hasten and short-cut the process. (Informant no. 03, 07, 09)

## 4. Discussion

This research was conducted with organizations in the service industry in Thailand that provide various types of services. Mixed-method research of documentary research and semi-structured interview for evaluating and exploring the process of using data analytics in the process of service innovation development.

The result of the process for the adoption of big-data analytics needs identification, and opportunity identification for service innovation found that the initial process focusing on data analysis to meet business needs is specific on a particular matter, despite the fact that data analytics processes for service innovation development have steps, as follows.

Business Objective and Requirement setting is the starting point for opportunity identification for Service Innovation. Research results showed that opportunity identification by setting objectives and goals for listening to voice of customers with specific listening channel and method based on customer-centric focus is to ensure that the listening covers all matters for value creation according to the study of Wirth and Hipp (2000) on data mining, which identifies on understanding the problems and converting them into requirements for data analytics that is to determine the type of specific data for collection (Wirth and Hipp 2000).

It is found that data identification and data acquisition are the objective conversion in step one to identify details of data collection method, data source, and data type for technically processing to find the linkage of customer experience in all interacting channels, especially those containing extensive data such as digital channel, according to research focusing on data for data analytics, which is the essential standard process for required data (Labrinidis and Jagadish 2012; Sumathy and Chidambaram 2013; Gandomi and Haider 2015; Dang and Ahmad 2014; Lehrer et al. 2018; Lim et al. 2018).

Data Understanding and Preparation is the following process of data collection, which is Data construction and transformation for analysis and initial study for data understanding. The initial

research is to describe the data characteristics by using descriptive statistics, which is briefly mentioned in previous research that the data preparation and understanding is one of the processes to gain benefits from data application (Wirth and Hipp 2000; Gandomi and Haider 2015).

Identifying the problem and customer insight is the new process from the in-depth interviews, which is the crucial process for focusing on data analytics for opportunity identification in service innovation development. It is necessary to work collaboratively between Data scientist and Domain Expert, who has front-line interaction with customer directly, to identify and reflect problems or true demands of customers, as well as analyzing data to create a prototype model prior to proceeding to the next steps, which is the integration of the qualitative data analytics from brainstorming and quantitative data analytics in accordance with research on New Service Development (NSD), which stated that Cross-Functional working and collaboration from front-line staffs are one of the important factors in the service system for the success of service innovation creation (Johnson 2000; Schneider and Bowen 1984).

In the process of Analysis and Modeling, it is found that using the initial model for deep analytics to identify form and relationship is the statistical analysis at a higher level for model creation. There are multiple analysis methods, for example, Association rule, Classification. This process relates to Data analytics research specifying the method of the analytics for Model creation (Wirth and Hipp 2000; Gandomi and Haider 2015; Sumathy and Chidambaram 2013; Dang and Ahmad 2014).

The model evaluation is to identify its quality level, whether meeting the objectives, and data credibility. If there are discrepancies, the model is required to be adjusted on its accuracy and reliability to the applicable level standard. The result of the study is in line with the findings indicating that data accuracy validation is the prior process to data application, and data credibility evaluation is based on confidence level in the statistical analysis (Wirth and Hipp 2000; Sumathy and Chidambaram 2013).

The last step is to use the result for practicing and applying into new service innovation processes, which is to extract the knowledge from big-data analytics for utilizing in New Service Development (NSD), which is considered as the front end of service development planning. For this research, this is the last process which is the starting point for New Service Development (NSD) for creating Service Innovation in the next process according to the research of Scheuing and Johnson (1989), who believed that the development of new services was not caused by occasion or situation; however, it was created from the designing processes and systematic methods, especially in the fuzzy front-end of service development, which is necessary to find insights and true demands (Scheuing and Johnson 1989; Johnson 2000).

As a guideline for the completeness of future research in the adoption of big data analytics for service innovation, action research should be conducted in order to obtain feedback from participants. The action research interconnects typical research to applied research for examining the adoption process of Big-data analytics intended for practical service innovation development and success factors together with the appropriate process adjustment. The action research has a systematic approach consisting of four procedures, which are (1) Plan, (2) Act, (3) Observe, and (4) Reflect and Re-plan to execute the cycle for the objective achievement. The result of the action research can be used to further improve the processes and procedures in a broader context.

## 5. Conclusions

This research shows that adopting big-data analytics for service innovation development is useful and valuable to the organization in order to develop service business capability to be at its forefront level, especially in the circumstance when there is a rapid transformation in customer behavior and technology in the digital economy these days. There are many forms to manage service innovation development. Many organizations usually perform using the inside out approach, which management would like to have speedy development of their service to serve market needs, with an immediate outcome. They generally gather data from within and incomplete implementation for data analytics adaptation. As a result, Thailand is lacking a systematic adoption of big-data analytics for service

innovation development, even though each service industry provides service via digital channels containing massive data. Additionally, there are limitations in adopting service innovation, data management, and knowledge extraction from data in order to develop service innovation, causing an unsuccessful achievement for service innovation or consuming a lot of time and budget to manage.

Results of this research show procedures and processes for adopting big-data analytics for service innovation development consisting of 7 steps; from setting business objectives to adopting the analytics; (1) Setting Business Objective, and Requirement, (2) Data Acquisition, (3) Data Understanding and Preparation, (4) Identifying the problem and customer insight, (5) Analysis and Modeling, (6) Evaluation, (7) Applying the results to the new service development process. In addition to that, this research indicates key success factors for adapt big-data analytics in Service innovation development, e.g., Organizational Capabilities, Resource in task domain, Data Security and Data Privacy, Data Management, and Innovation Management, Managing Management expectation to be a guideline practice for service industry to drive their business to its forefront level and create competitive advantage in the digital economy era.

**Author Contributions:** This study has been supervised by T.T., A.C. and S.T. As for the research conducted and reported by N.T. All authors contributed to develop, review and revise the manuscript. All authors have read and agreed to the published version of the manuscript.

**Funding:** This research received no external funding.

**Conflicts of Interest:** The authors declare no conflict of interest.

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
