# Peer review of "Adopt Big-Data Analytics to Explore and Exploit the New Value for Service Innovation"

_socsci, doi:10.3390/socsci9030029_

Round 1

Reviewer 1 Report

The abstract mentions scientific methods used in the research. "Semi-structured interview in depth". The same is repeated in the methodological part. As generally accepted typolygy of interviews suggests three types of interviews could be applied: "structured interview", "semi-structured interview", "interview in depth". They are presented as different scientific methods with a different types of instruments and procedures of its application. As interviews in depth are having a tool as a list of topics to be discussed and no specific questions included (questions are based on responses collected), it seems that authors used the method of semi-structured interview (particular questionaire (with small changes in everycase) even with open ended questions). Therefore, it is recommended to authors to clarify which scientific method was used in their research to avoid any misundertandings. 

There is the lack of explanation why such number of interviews were included to the sample. How it represents the whole situation in Thailand business sector? What were criteria for sample selection (not only persons, but companies too)?

Methodological part also presents that "The rights of the sampling group were protected", "In order to protect the privacy rights of key informants, the unique code is set for summarizing the conclusion and analyzing data for the report". However, confidentiality as a principle of research ethics has been violated as authors have give the Table 2 with names of interviewees and companies connected to codes of informants. There is nothing added to the research ethics about permissions in writing got from interviewees to use their names openly. Therefore, authors must do needed corrections - if permissions were got, then to add this information in research ethics' description, if not - to eliminate or adjust the Table 2 and/or just give the list of companies, positions, average experience in years in the text in a different order to avoid any disclosure of personalities.

References in the text are given, however the style of their presentation could be improved: references could be given fully in brackets, and just ones, with the purpose to emphasize authors, could be given in the text.

Answers from informants should be given in a different shrift (i.e. Italic) to be identified easier.

Codes of Interviewees differ in the Table 3 and other text. They should be the same to disclose the position of particular intreviewee.

Besides terminology differs in naming the sample: experts, informants, interviewees. No clarification why it is.

It is a bit strange coding sytem: "Key informant no.,01,05,07,08,11". If 5 different interviewees have emphasized the same approach or aspect why authors name them "key informant"?

The suggestion about the action research in the future research is not detailed and explained enough.

Author Response

Thank you for positive and constructive suggestions. The manuscript was significantly improved after revised as your recommendation.

Reviewer 2 Report

The paper is interesting especially for the geographical context analysed and the research method used (and clearly described).

I suggest you to replace the out-of-date concept of consumer (from introduction to paragraph 2 and in the conclusions) with "customer" as service literature widely prefer and use this one. 

Finally, in some points, the punctuation is used incorrectly (e.g.: p. 1, last paragraph, colon "(WIPO): a specialized agency of United Nations (UN)").

Author Response

Thank you for the opportunity to revise the manuscript and would like to express our thanks to you very much for the positive feedback. 

Reviewer 3 Report

This paper described the results of the qualitative research on big-data utilization in the business field in Thailand, by the semi-structured interviews and literature reviews. The authors set two research questions; one is the question about conducting a big-data analytics process, needs exploration, and opportunity identification for service innovation development and another is what factors impact the success of applying big-data analysis.

They summarized the results of interviews into seven steps for the first question and five issues for the second question. However, it is an abstract description and lacks a persuasive explanation. That is, these items described in this paper have already been widely common and there are no novel findings.

For example, the paper described step two as follows:

To listen to customer feedback (Voice of Customer) and to identify customer’s important information such as what information and which channel related to and having relationship with customer experience, including clearly indicate the source of information. After that define the definition for each type of data (Data field description) and specific data gathering.

However, readers would like to know more practical findings, such as how to listen to customer feedback, how to identify customer's important information, and so on. Was there such information in the comments from the key informants?

Followings are editorial issues:

  • The section four is not mentioning the discussion. It denotes the result of the literature review. Therefore, the section title should be changed.
  • There are several unnatural expressions (and grammatical errors). It would be better to be checked by native English speakers.

Author Response

Thank you for the recommendation, The manuscript is substantially improved after revising.

Round 2

Reviewer 3 Report

I confirmed that the quality of this paper was improved.